# Exploration of the Design of Spiderweb-Inspired Structures for Vibration-Driven Sensing

**DOI:** 10.3390/biomimetics8010111

**Published:** 2023-03-08

**Authors:** Mahdi Naderinejad, Kai Junge, Josie Hughes

**Affiliations:** 1Department of Mechanical Engineering, College of Engineering, University of Tehran, Tehran 1417935840, Iran; 2CREATE Lab, IGM, STI, EPFL, 1015 Lausanne, Switzerland

**Keywords:** tactile sensing, morphological computation, soft robotics

## Abstract

In the quest to develop large-area soft sensors, we can look to nature for many examples. Spiderwebs show many fascinating properties that we can seek to understand and replicate in order to develop large-area, soft, and deformable sensing structures. Spiders’ webs are used not only to capture prey, but also to localize their prey through the vibrations that they feel through their legs. Inspired by spiderwebs, we developed a large-area tactile sensor for localizing contact points through vibration sensing. We hypothesize that the structure of a web can be leveraged to amplify, filter, or otherwise morphologically tune vibrations to improve sensing capabilities. To explore this design space, we created a means of computationally designing and 3D printing web structures. By using vibration sensors mounted on the edges of webs to simulate a spider monitoring vibrations, we show how varying the structural properties affects the localization performance when using vibration sensors and long short-term memory (LSTM)-based neural network classifiers. We seek to explain the classification performance seen in different webs by considering various metrics of information content for different webs and, hence, provide insight into how bio-inspired spiderwebs can be used to assist large-area sensing structures.

## 1. Introduction

Tactile sensing over large and soft or deformable areas is challenging due to the many degrees of freedom of these structures [1]. There is much work exploring the development of novel materials [2] and scalable approaches to measuring signals with minimal electrodes to cover large regions [3,4]. However, nature provides many examples of solutions to this problem. One specific example of robust sensing over a large area is that of spiders and the use of their webs. Spiders are able to create their web structures from silk and then detect vibrations and deformations through a specialized organ, the slit sensilla [5,6]. When insects land on the web, the vibrations alone allow the spider to locate its prey and move to it. They can also sense damage or potential mates through vibration sensing with their webs [7,8]. The web’s structure and materials, along with the spider’s vibration receptors, enable tactile sensing and localization via vibrational transduction [9,10]. This provides valuable inspiration for the development of large-area soft tactile sensors through deformable ‘web’ structures with vibration receptors. The building, functionality, and evolution of spiderwebs are complex, and they have long been studied by biologists [6]. Their morphology and function can vary significantly depending on the species of spider and the required functionality. Web structures have been shown to vary as spiders mature [11], and spiders have been shown to selectively adjust the mechanical properties of their webs to aid in their functionality. We specifically took inspiration from one specific design of webs, the orb web [12], which was characterized as a 2D structure constructed from a circular net with numerous radial and spiral threads.

A spiderweb is not solely a passive mechanism for trapping prey; rather, it actively contributes to the localization of interactions through the vibrations received by the spider, which is usually located at the center of the web [13]. Another property of spiderwebs is their tolerance to damage—they can retain their structural properties and functions despite damage to parts of the structure [14]. This might also extend to their sensing properties. Previous works have explored the mechanical properties of spiderweb morphologies [15] while utilizing a variety of different fabrication and simulation methods [16]. Analysis of vibration signals showed that a web’s geometry and dynamics can improve the localization of prey through impact vibrations [10]. Furthermore, by 3D printing spiderwebs, their nonlinear properties have been highlighted, their dynamic effects have been explored [17], and their potential to act as a physical reservoir for computation has been proposed [15]. The development of soft sensors inspired by web structures has straddled a range of different fabrication and sensing modalities. This includes web-inspired graphene tactile sensors [18], flexible sensors for simultaneous pressure and strain detection [19], and laser-induced graphene stress sensors in web structures [20]. This body of work highlights the advantages of web morphologies. However, existing approaches largely exploit material properties for sensing, as opposed to vibrations. In addition, the morphology of such sensors has not been explored in depth.

In addition to web-inspired sensing structures, there has been a significant amount of work exploring vibrations in tactile sensing, including vibration sensing in fingertips [21] and acoustic sensing through microphones embedded in pneumatic fingers [22]. Studies such as [23,24,25] also actively used vibrations for tactile sensing, but they limited the sensing area to the ‘tip’ of the device. The use of accelerometers for large-scale tactile sensing was demonstrated in works such as [26] to expand the tactile feedback on a PR2 robot, as well as in [27] to develop a sensor for performing the localization of a sliding contact over a silicone patch, but in both cases, the effects of variations in the sensor morphology were not explored.

Building upon this existing body of work, which has demonstrated the advantages of the ‘morphological computation’ of web structures, we propose a new sensing approach that combines deformable web structures with accelerometers to enable the localization of contact through vibration monitoring. To better understand the role of webs, we explore the extent to which changes in web-inspired morphologies affect the localization accuracy and the nature of the data created in terms of the information content (entropy), reproducibility, and similarity across the web (joint entropy).

Inspired by the structures made by orb spiders, we developed and sensorized webs with different geometrical designs. The webs were fabricated by using FDM 3D printing to create thin deformable PLA structures. A custom indentation testing setup was developed to automatically make single points of contact on the webs. Using this setup, we were able to generate a large dataset of indentations at different locations.

Using this dataset we leveraged time-series-based neural network classification with a long short-term memory (LSTM) network to perform classification of the location of contact. We propose the use of metrics from information theory as a means of understanding or exploring these results to comment on how different morphological parameters in a web affect its localization performance. This approach is summarized in Figure 1.

In the remainder of this paper, we first present the methods used developed to fabricate, test, and evaluate the localization performance of four differently shaped webs. This is followed by the experimental results and a conclusion. We show how, with three accelerometers on the highest-performing web (among the four webs), a localization performance of over 95% could be achieved for 31 different location classes. Although only four webs were studied, the variation across these led to significant variations in the accuracy and information content metrics, demonstrating the importance of the morphology of the web structure.

## 2. Materials and Methods

In this section, we detail the fabrication of the webs, the integration of sensing, and the training implemented for the classification of different locations of contact. With these methods, we explore the effects on the accuracy of localization when using different web morphologies and different numbers of accelerometers.

### 2.1. Web Design

Previous work has identified the ‘construction rules’ of real spiders and has used these to computationally develop plausible web structures [28,29]. We reduced the complexity of the webs by regrouping and simplifying typical structures found in webs, e.g., the hub, mooring, frame, radials, etc. The design of our spirals resembled a temporary spiral, which is a structure made by a spider that stabilizes its web, allows the spider to cross between radii as it continues to build, and guides the placement of the final capture spiral. The growth of the radial spacing of the spiral was set to be logarithmic, with a higher density of ‘inner’ spirals. The **numbers** of radii (*r*) and spirals (*s*) could be varied, with the length of the radii being fixed at 50 mm. The angle between the radii $\theta_{r}$ determined the spacing, and this was given by $\theta_{r}=2\pi /r\pm rand\left(0,x\right)$, where a uniform random number generator with a range between 0 and π/6 was used to simulate webs with fixed or random variable angles. By setting these rules for any values of $r,s,\theta_{r}$, the web morphology could be generated. Although there are many other parameters to investigate, these were chosen because they were found to be key features in spiderwebs.

By using OpenSCAD, a programmatic and parametric CAD software, web structures with varying radii, spirals, and angles could be designed and then output as an .stl file. The thickness of each strand of the web was chosen to match the diameter of the 3D printer extruder (0.6 mm) such that a single thread was extruded.

### 2.2. Fabrication and Sensing

The webs were printed in PLA by using a desktop FDM printer (CREALITY CR20) with a nozzle diameter of 0.6 mm. To achieve a compliant surface, each web structure was a single extrusion (width: 0.6 mm). The variance in the diameter of the extrusion was reported to be ±0.02 mm for the printing technique used. Each web was fabricated as a flat structure (with tabs added for the accelerometers) on the print bed. After printing, they were removed and attached to a stiff outer ring that was raised up on extrusions such that it could be deformed without contact with the surroundings. Being able to 3D print the webs allowed for rapid exploration with uniform material properties. After fabrication of the 3D-printed web structure, accelerometers were attached at approximately equal intervals around the web in a given orientation and at a fixed distance (10 mm) from the outer hub of the web. Although spiders typically sense prey from the hub area of the web, we placed the sensors at the extremes of the radii so that they did not affect the flexibility of the web structure and so that the wiring was at the edges, which is advantageous for many applications.

To allow for the exploration of how the geometry of the web affected the classification performance, four different geometries were tested. Their geometric parameters are summarized in Table 1, and they are visually shown in Figure 2. Here, we had a base web structure and independently varied the design parameters. Figure 2 shows the variety of the structures, as well as the mounting points for the accelerometers. Figure 3 shows the overall pipeline which combines the parametric design, fast fabrication and the experimental data collection detailed in the next section.

### 2.3. Experimental Data Collection

The experimental setup needed to have the capability of providing an impulse to the spiderweb in arbitrary locations while collecting accelerometer data. To collect a large enough dataset, this process needed to be performed in an autonomous fashion. Figure 4 shows this experiment setup, which comprised a gantry from a repurposed single-axis FDM 3D printer. The hot end of the printer was replaced with a solenoid that could be activated with a microcontroller (Arduino UNO) to generate an impulse force that acted on the spiderweb. The solenoid had a stroke of 3 mm and was positioned 1 mm above the web, such that there were 2 mm of deformation when extended. On the end of the solenoid, there was a 3D-printed cone structure that had a radius of 7.5 mm and a height of 2 mm. The shape of this indenter was kept constant.

The Cartesian platform of the 3D printer could be easily and directly controlled through G-code. Furthermore, the G-code commands, solenoid impulse commands, and accelerometer data collection are synchronized, allowing for flexible and automatic data collection by the system. One or more accelerometers could be placed on the spiderweb and read by another microcontroller (Teensy 4.0). Three-axis 3g accelerometers (Adafruit LIS3DH Triple-Axis Accelerometers) were mounted on the webs with an orientation in which the z-axis was aligned with gravity and the x-axis was aligned with the nearest edge of the web. The readout was gravity compensated, and the accelerometers were sampled over I2C at approximately 5 kHz.

An example of the response from a single accelerometer on web 1 to the lowering and raising of the solenoid indenter is shown in Figure 5. The image depicts the distortion that occurred in the web, which corresponded to a vibration pattern observed by an accelerometer. This pattern arose from both the expansion and the contraction of the indentation. Although the patterns were largely similar for all three axes of the accelerometer response, the decay rate and magnitude showed variations.

### 2.4. Localization Estimation

To measure the localization accuracy of the webs, we explored the ability to correctly classify the responses from the accelerometers for 31 classes across the surfaces of the webs. Each class corresponded to a fixed location on all webs. These locations started from the center of the web and spiraled outwards. Thus, the performance indicator that was measured was the accuracy in assigning a recorded indentation to the correct location class. To generate training data, the printer platform moved to these points at a fixed height from the web surface, and the solenoid was rapidly activated to perform an impulse-based interaction with the web. Each point was tested 20 times in random order, with sufficient time left between interactions to allow vibrations to settle. To avoid overfitting, some random noise was introduced into the locations.

To classify the particular point of contact, a bidirectional LSTM was used to train the network. An LSTM utilizes past states to make predictions [30]. This is a neural network architecture that is increasingly being used for tactile sensor classification and detection [31,32]. To train the LSTM network with the acceleration data, the collected dataset was split in a ratio of 70:30, meaning that 70% of the available time series were used for training, and the rest were used for validation. To classify the outcomes into 31 classes (expressing 31 poked points), the LSTM network was used. This contained one bidirectional layer, one fully connected layer, and one softmax layer; it was trained with 70–100 hidden units and a maximum of 80 epochs. The learning rate was set to 0.001, and the ‘adam’ solver was used by the network. Each data sample was preprocessed by normalizing to the maximum response of one accelerometer (A1) so that the trained network would be robust to various indentation forces and sizes.

### 2.5. Information Theory Metrics

To interpret the varying abilities of the different webs, we propose the use of various means of analyzing and computing the information content gained from each web. These methods are interesting, as they are independent of the physical meaning coming from physical data collection. This allowed us to provide some analysis or intuition of how the webs filtered or ‘shaped’ the information, which, in turn, assisted with the learning-based classification. The information-based metrics that we considered included:**Euclidean distance after dynamic time warping (DTW).** Dynamic time warping is a time-series analysis that aligns and warps two time series to achieve the best match [33]. After performing the time warping to align and ‘stretch’ two time series, the Euclidean distance between these two aligned time series could then be determined as a measure of closeness or similarity. For each web and each point, we computed the Euclidean distance after time warping for all repeats to obtain a quantitative metric of the similarity or repeatability of the time series that were generated. DTW ensured that the time series were appropriately aligned. We used this to assess how the web structure could affect the repeatability or reliability of the response.**Entropy.** This provides a measure of the amount of information held in data [34]. For each web, we computed the average entropy across all samples to determine how the web structure affected the web’s information content.**Joint entropy.** This metric provides a measure of the uncertainty between two random variables. This can be used in the context of time series to understand the amount of certainty or the lack of mutual information between two time series. The higher the joint entropy is, the lower the mutual information or the mutual entropy between two variables or time series will be. For each web, we computed the average joint entropy between all possible pairs of points. This is a metric that was previously explored as a means of optimizing sensor morphology [35].

## 3. Results

### 3.1. Exemplary Response

An exemplary response in the z-direction (upward-facing direction) from the three accelerometers mounted on Web 1 when probed in two different locations is shown in Figure 6. This demonstrates how the three accelerometers each provided a different response for the same location of contact. In addition, when the location was changed, there was a distinct difference in the response, with the relative magnitudes of the three responses shifting and the frequency component changing. This demonstrated how the responses from the accelerometers changed with the distance, as well as how the three accelerometers received different vibrations.

### 3.2. Classification Accuracy

The classification success of the LSTMs were found for all webs when using one, two, and three accelerometers for the training dataset that was previously described. For each web and each test point, the accuracy (average across all test data) could be determined. This is first shown graphically, overlaid over the structure of the web, with the classification accuracy at each point being shown (Figure 7). To allow for a more quantitative analysis, boxplots showing the classification accuracy of each web for one, two, and three accelerometers is given in Figure 8.

For localization with three accelerometers, Web 1, which had the fewest spirals and radii, as well as regular angles, performed the best. This was followed by Web 4 (which had more radii), Web 3 (more spirals), and Web 2 (non-uniform spacing). As shown in Figure 7, for the best-performing webs (Webs 1 and 4) and with three accelerometers, the accuracy was approximately constant across the web structure. For Web 3, there was lower accuracy in the center, where the LSTM showed lower localization performance. This could have been due to the extremely close spacing of the web structure, resulting in points with similar temporal responses.

Considering the accuracy of localizing with only one or two accelerometers, it could be seen that the points close to the sensor tended to have a higher localization accuracy (as shown in Figure 7). This was reflected in the overall accuracy, with the accuracy dropping for all webs from over 95% to around 90% with two accelerometers. However, for reconstruction with one accelerometer, we saw a far larger drop in accuracy, except in the case of Web 3, which showed significantly better performance. In this case, although the close packing of the spirals did not help in the localization between close points in the center, it appeared to assist with localization over the entire web structure.

In all cases, introducing non-uniform spacing between the radii did not appear to assist in localization. Furthermore, having fewer web structures (e.g., spirals and radii) appeared to aid in reconstruction with multiple accelerometers, but when there was only one accelerometer, having a tighter spiral was advantageous.

### 3.3. Information Content Metrics

To explore why certain web structures aided in localization, the different information metrics identified in Section 3.2 were computed from the raw data collected for each web. Figure 9 shows the average reconstruction for each web with different numbers of accelerometers alongside the three metrics (DTW, entropy, joint entropy), which were computed only for the three-accelerometer dataset. As reported previously, Web 1 performed the best, followed by Web 4, Web 3, and Web 2. Considering the results, when the Euclidean distance after DTW was used as a measure of similarity between repeats, it was approximately constant among all webs, although it was lower for Web 3, which had a greater number of spirals. This perhaps reflected the larger area coverage of the webs and the higher repeatability in their responses. However, there was no clear correspondence between DTW and localization accuracy.

Concerning the entropy and joint entropy, we saw a clear relationship between the joint entropy and the reconstruction accuracy, with the web with the highest joint entropy having the highest reconstruction accuracy. This suggests that each point’s location on the web structure had the least shared information, which could have contributed to this higher reconstruction accuracy. The joint entropy was significantly lower for the worst-performing web (Web 2). For the entropy, we saw that the worst-performing web (Web 2) had the highest entropy, and the better-performing webs had a lower entropy.

Although these information theory metrics cannot fully explain the performance of the LSTM-based classifier, they go some way in explaining how an analysis of the raw data from the sensors can be used to predict or understand the classification and localization performance.

## 4. Conclusions

Spiderwebs show some fascinating properties in terms of large-area sensing due to spiders’ abilities to use vibrations from environmental interferences. Their morphological properties provide many insights into the design and development of structures that can be used for large-area tactile sensing. In this paper, we introduce a means of rapidly fabricating spiderwebs with variable spiderweb-inspired geometries. By using accelerometers, we can capture vibrations that form on these web structures when an impulsive force is applied. We show that the morphology affects the ability to localize an impact, and the web performance is linked to the joint entropy of the raw data.

Going forward, there is much to explore in order to build upon this initial exploration and to further leverage the ability to rapidly fabricate these webs in a computational manner. Firstly, exploring more webs with increasingly complex structures would allow this hypothesis and approach to be further verified. This could aid in conclusively demonstrating if the higher performance can be explained by the information content metrics. Benchmarking against non-bio-inspired structures, such as grids, could also help in benchmarking the capabilities of these web structures. Inspired by the damage-resistant nature of spiderwebs, the ability to localize after damage would be an additional direction to explore. A second avenue of further work is the extension of the topological optimization of web structures by leveraging methods through iterative real-world testing and evaluation. Although these structures are challenging to model, investigating their performance by using FEA/FEM analysis could also be beneficial in the computational optimization of webs. This would also allow for the exploration of how the scale of the structure and the material properties affect the localization capabilities, which could allow for further optimization of the structures of webs. Alongside the optimization of the shape, the effects of structural failures in the form of removing/breaking web links on the localization accuracy can lead to an understanding of the resilience of this type of sensor. It would also be interesting to explore capturing signals at a higher sampling rate to allow for analysis across a larger bandwidth. This could be performed through the use of transducers such as vibrometers.

The ability to place accelerometers on the edge of a structure to localize deformation is attractive for many applications, as the structure is not affected. In addition, the proposed approach could be highly scalable, as a finite number of accelerometers are required. These advantages make this technique well suited for applications such as forming large-area sensorized surfaces for a robot palm, or even for forming deformable feet for a soft robot to detect contact or surface types.

## Figures and Tables

**Figure 1 biomimetics-08-00111-f001:**
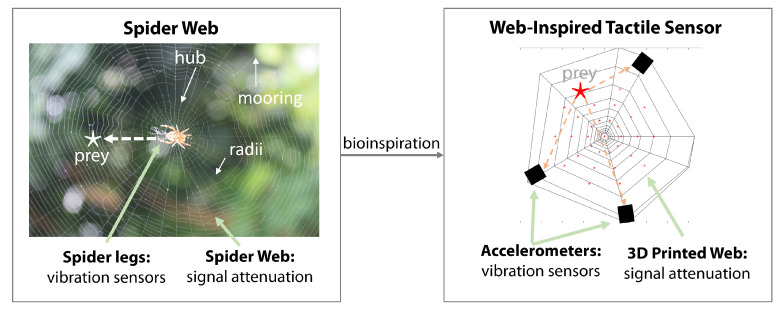
Summary of our bio-inspired approach of using accelerometers to identify the location of a contact, which was inspired by spiders and their webs.

**Figure 2 biomimetics-08-00111-f002:**
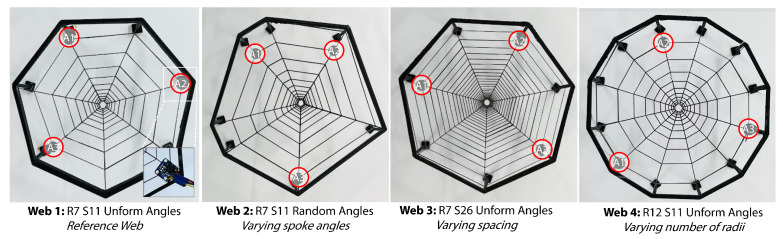
Images of the four webs created showing the web structure and the mounting points of the accelerometers (shown in the red circles).

**Figure 3 biomimetics-08-00111-f003:**
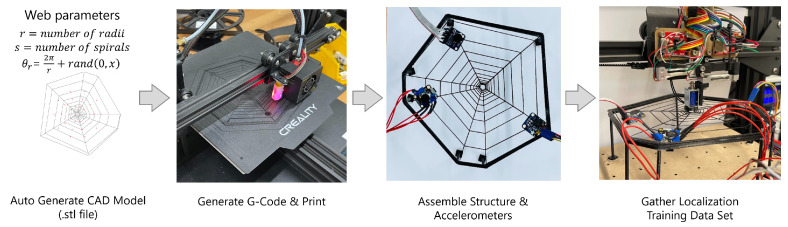
Summary of pipeline. The design was generated and printed, the accelerometers were assembled, and the test data were gathered by using the experimental setup.

**Figure 4 biomimetics-08-00111-f004:**
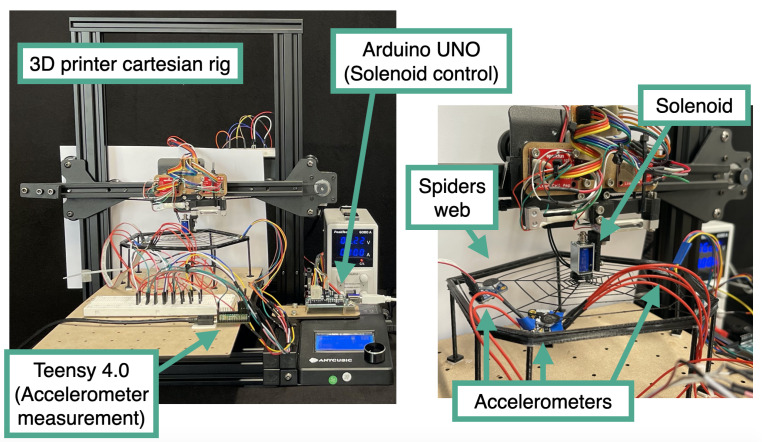
Automatic data collection setup with a Cartesian 3D printer rig. Right: Full image of the setup. Left: Closeup of the spiderweb with accelerometers attached and the impulse solenoid.

**Figure 5 biomimetics-08-00111-f005:**
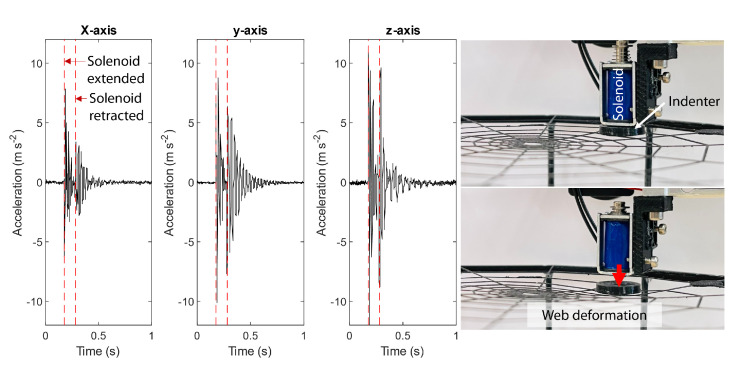
Visualization of the collected data and the deformation caused by the indenter mounted on the hot end of the 3D printer. The indentation time was 0.15 s, and the release time was 0.25 s.

**Figure 6 biomimetics-08-00111-f006:**
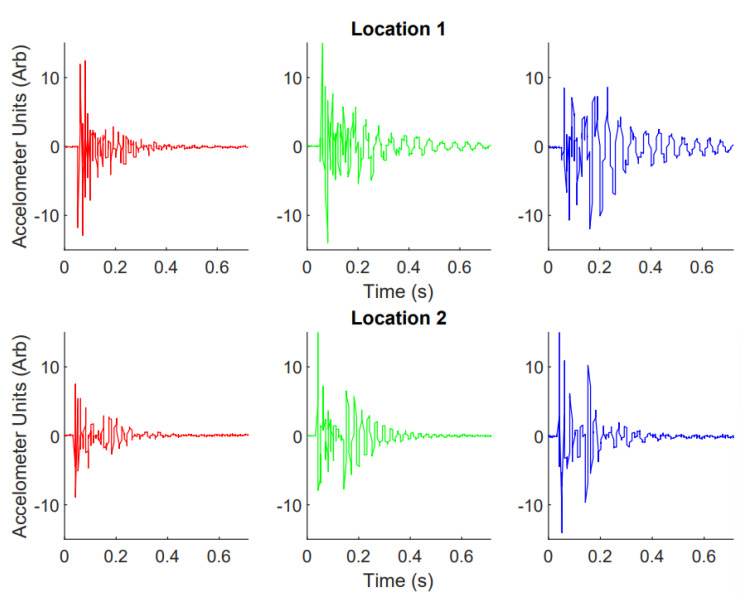
Example of the data collected when the indenter on the end of the 3D printer setup is lowered and raised. Location 1. This shows data from *Web 1* with a single accelerometer, corresponding in the setup in Figure 7 (Web 1, one accelerometer). Location 1 corresponds to Point 10 (in Figure 7), and Location 2 corresponds Point 25.

**Figure 7 biomimetics-08-00111-f007:**
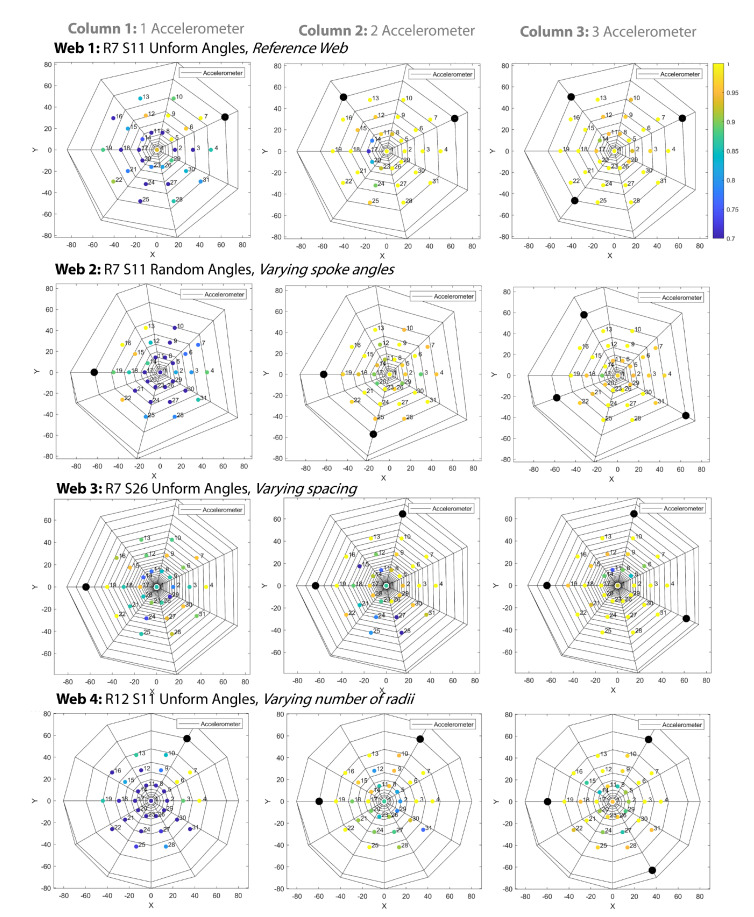
The classification accuracy is plotted for the four different webs structures (each row) with one, two, and three accelerometers on the web structure for sensing (each column). The colored bar indicates the classification accuracy, and the black dots indicate the location of the accelerometers.

**Figure 8 biomimetics-08-00111-f008:**
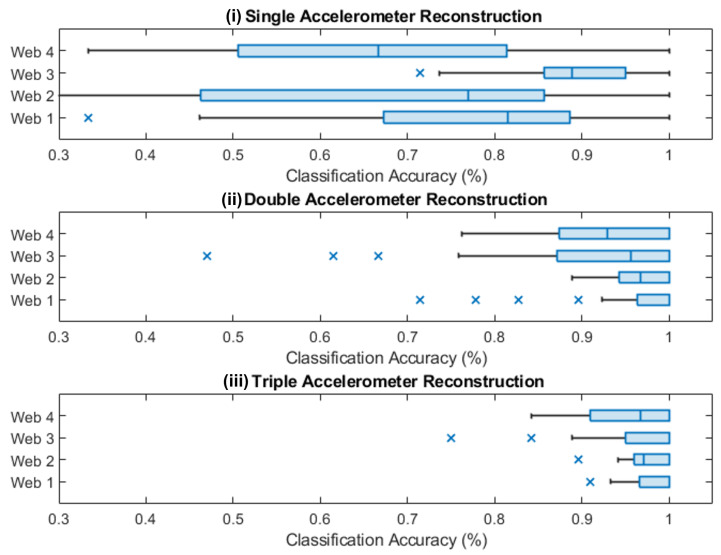
Boxplots representing the classification accuracy for each web for one, two and three accelereometers. (**i**) Shows a box plot for only a sigle accelerometer, (**ii**) for two accelerometers and (**iii**) for three accelerometers.

**Figure 9 biomimetics-08-00111-f009:**
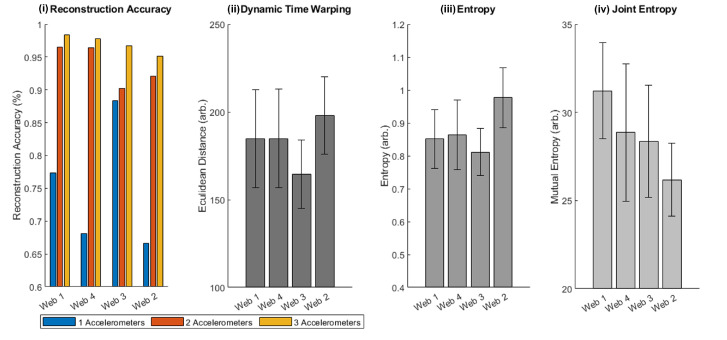
For each of the four webs, the figure reports (**i**) the average reconstruction accuracy for the different webs and with different numbers of accelerometers, (**ii**) error after dynamic time warping for the four webs, (**iii**) entropy for the four webs, and (**iv**) joint entropy between different locations on the four different webs.

**Table 1 biomimetics-08-00111-t001:** Summary of the geometries of the four different webs that were tested.

	Web 1	Web 2	Web 3	Web 4
Number of Radii (r)	7	7	7	12
Spirals (s)	11	11	26	11
Angles $\left(\theta_{r}\right)$	Uniform	Random	Uniform	Uniform
Number of Sensors	3	3	3	3

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
