# Peer review of "Exploration of the Design of Spiderweb-Inspired Structures for Vibration-Driven Sensing"

_biomimetics, 2023, doi:10.3390/biomimetics8010111_

Round 1

Reviewer 1 Report

In this manuscript, the authors reported a design means of web structure inspired by spider web structure, and it can be combined with accelerometers to realize the vibration recognition from different positions through the neural network. However, it seems less innovative of the computational design means, information content metrics, and neural network, and the work exploring how the structure of proposed webs contributes to vibration sensing needs great improvement. 

 1.     The design and fabrication of the web structures, neural network algorithm, and index lack of innovation. The authors need to clarify the specific problems or reasons that lead the author to carry out the research in the paper in the introduction part. In other words, the authors should explain what deficiencies in the current study of spider webs prompted this work.

 2.     The proposed web structure has three main structural parameters: maximum radius, spiral number, and radial line distribution. But only four webs are designed and studied. More webs with various radius and spiral numbers, and uniform or random radial line distribution should be proposed and studied to improve the persuasiveness and credibility of the conclusion.

 3.     As described in the manuscript, the bottom end of the pulse loading device is circular. When the pulse was loaded, the contact position with the web structure is not the center of the loading device. Whether the size of the loading device has a great impact on recognition accuracy? Further explanation is required. Moreover, it seems contact region recognition is more reasonable than the recognition of specific points.

 4.     More details about how to convert the collected vibration signals into the three theoretical metrics used in neural networks need to be explained for readers’ comprehension.

 5.     It can be seen from the results that the recognition accuracy has been greatly improved with the increase of accelerometer numbers. Is the identification accuracy highest when the number of accelerometers is the same as the number of radial network lines? The author needs to analyze the relationship between the network structure parameters, the number of accelerometers, and the recognition accuracy to improve this work.

 6.     Minor suggestions: Add more description information in the figure, such as loading and unloading times in Figure.

Reviewer 2 Report

The only relevant field in which I have expertise is the biology of spider webs, so I will limit my comments to this topic. All of my comments below on spider webs are based on a recent, extensive review of spider web biology: W. Eberhard 2020. Spider webs behavior, function and evolution. Univ. Chicago Press. This book synthesizes literature (most not cited in the current ms); its pages 126-129 are especially useful for the topics in the current paper. I believe that the paper under review would benefit from including information in this and other parts of the book. I am, however, the author of this book, and I believe it is unacceptable for a reviewer to require that authors use his own work. I thus only recommend its inclusion in this ms.

Throughout, the authors need to clarify that they are not talking about spider webs in general, but only a specific design of spider web – the orb web. There are many other spider web designs (which might or might not provide inspiration for mimetic human structures).

The use of spider webs as inspiration for man-made objects needs to include considerations of scale. Specifically, a line can vibrate along the axis of the line (longitudinal vibrations) and perpendicular to the line (transverse vibrations). Because the lines in an orb web are very thin and light, friction with the air is expected to rapidly damp their transverse vibrations; several lines of evidence indicate that the biologically important vibrations in orb webs are longitudinal. In most man-made structures (including the model webs discussed in this paper), in contrast, the balance between weight and friction with the surrounding medium may be quite different (an exception would be structures under water). I am not sure what effect these scale-related factors will have on the data and arguments in this paper. I believe, from looking at Figs. 4 and 5), that they measured only transverse vibrations. And I believe that transmission from one line to another of longitudinal vibrations is much more dependent on the angle made between the two lines than is transmission of transverse vibrations (longitudinal vibrations are poorly transferred when the angle is 90o between lines, much better transmitted when the angle is close to 180o; transmission of transverse vibrations is less dependent on the angle). In sum, I believe the authors need to discuss this scaling factor specifically.

Perhaps my largest doubt concerns claims that “The spider web is not just a passive trap but has also an important active role contributing to the localization of interactions by filtering the vibrations, showing morphological computation.” This contrasts with the conclusion from data on webs and prey summarized in the book that “It is unlikely that different web designs are ‘tuned’ to transmit the vibrations of certain preferred prey more effectively.”  

The statement that the temporary spiral (or the sticky spiral for that matter) is “defined to be logarithmic in nature” is far from correct, at least as far as real orb webs are concerned (documentation in the book). This has led the authors to use model webs with non-spiderlike patterns of spaces between loops of spiral. The pattern of variation in the spaces between successive loops of spiral in the different model webs in Fig. 2, with spaces showing a negative relation to the distance from the hub, does not precisely reflect patterns spider webs: a negative relation occurs in some portions of the orbs of some species, but not in other portions of the same orbs, or in the orbs of other species (documentation in the book)

One further difference with living spiders is that spiders perceive the location of the prey while resting at the hub of the web. The accelerometers in the experimental setup were on radii, far from the hub.

In sum, several statements concerning spider webs in this paper can be improved. It is not clear to me, however, that any of these imprecisions has any effect on the accelerometer data or the interpretations with respect to localization accuracy and subsequent information theory analyses. Their data would seem to be valid (as far as my expertise goes); my reservations simply introduce the question of how accurately their test webs have mimicked spider orb webs.

Reviewer 3 Report

This paper reports on the fabrication of sensored web structures, an experimental procedure to provide repeated contacts, a LSTM NN approach to estimate the point at which the web was contacted and some information technology analysis of the generated data as a way of assessing the quality of data generated by the system.   In general the introduction would benefit from a more in-depth analysis of the prior art. The design of the webs used seem somewhat arbitrary. A simulated response to refine the design may have been helpful. More detail on the computational approach taken especially around the Information theory assessment would be useful. There is no discussion about the application of the technology. Generally, the paper's language should be refined as there is some repetition while some parts are unclear and there are many typos.     More specific points are given below:   line 9: write LSTM as Long Short-Term Memory (LSTM)  

Line 17 unclear. What is read-out?

"However, nature provides many examples of these solution" rephrase   line 19  specializing -> specialized   line 20  is webs -> its webs "spans the environment" vague. rephrase   line 21. period would be better than semicolon   line 24-26 needs rephrasing   line 28 and -> with   line 34, has also -> also has comma after role   line 36 "they remain functional even after damage" superfluous or rephrase   line 41 ask -> act   Would have liked to see greater depth of analysis on previous research in this space. The analysis done on spider webs, their synthetic counterparts and research into their use as a sensing method. There seems to have been several publications in this space https://pubs.rsc.org/en/content/articlelanding/2018/cc/c8cc02339e/unauth https://pubs.acs.org/doi/10.1021/acsami.0c21960 https://www.mdpi.com/2079-6412/13/1/155   The section on tactile sensing, line 42-44 only gives a small number of examples and is phrased as if this encompasses the whole field. Maybe give further examples and point to a review paper as this is a highly researched area No real argument for the importance of this work, what applications etc.    line 47 localization of contact.-> perform localization of contact.   line 51 "to great a deformable" -> "to create deformable"   line 52 unclear what "indentations" are at this point. points of contact?   line 59 cannot claim optimal web. It was the best of 4 webs tested    line 66. A figure labelling the parts of the web listed would be useful.   line 71 "The spirals are defined to be logarithmic in nature, replicating spiders" needs clarification the spacing is logarithmic, or length or what exactly?   line 73 does r here refer to the number of radii? Not the radius length?  Should the random number vary  between +- x so the spacing can also be less than the even spacing?   line 76. No evidence given for why these parameters would be dominant.    line 80 "design" -> generated This section seems to be more of a random web generator than computational design. It seems that it would be prudent to do some sort of simulated web analysis to determine the important design properties of the web. Things like stiffness, elasticity and other material properties seem not to have been considered. It may also make it possible to assess if the approach could work at larger or smaller scales. Is there an advantage to a web over a solid sheet? Simulation would also guide the placement and number of sensors which seems to be a parameter that was not considered. Simulation would also allow more than 4 web designs to be considered.    Line 85 should specify the layer height or was it also 0.6mm? Could this be kept sufficiently consistent with an FDM printer?   Figure 3 includes equations with degrees but radians used elsewhere.    Should give the model of the accelerometers used.  What was the orientation of the sensors? The data shown is gravity compensated? Were other sensors such as strain sensors considered?   line 113 see->seen   If 20 tests were done at each point it would be interesting to assess the repeatability of the tests.    Why does figure 5 only show one accelerometer output if 3 were used?   By how much does the solenoid deform the web? was the diameter of the indenter varied?   line 115 accelerate -> acceleration   line 125 I don't think figure 3 shows this   line 127 missing reference   line 117 I don't think Reconstruction is a good name for this section. Maybe "Contact point estimation"?   when splitting the data set, as there are 20 samples at each point its likely you are training the data specifically at all the points you're then testing. Could this therefore be over trained? Should the validation data be at additional random points?   Figure 6. On what web is this data from, what contact locations, why are arbitrary units used? Why is only z axis shown if all showed similar accelerations? was only the z axis used in the analysis or perhaps the magnitude of the acceleration vector?   Section 3.2 It's unclear how accuracy is being expressed as a percentage here. Is it the distance between the estimated and actual contact point as a percentage of the web diameter? Or the number of estimated validation contacts within a certain distance of the actual contact point?  I think actually the LSTM only sorts into one of the 31 bins where data was collected. It seems like a large limitation if arbitrary contact points cannot be assessed     Up to this point it hasn't been made explicitly clear that the number of accelerometers necessary is to be assessed.    Figure 7 should say 1-3 accelerometers under the images. Also some alignment issues   Conclusion: Were the webs damaged during the process which could result in the behavior varying over the test? Would be good to discuss applications. What can this technology be applied to? The conclusions themselves should be expanded if possible. Can these results be compared to anything else in the literature? 

Round 2

Reviewer 1 Report

The author has addressed all the comments, so this paper can be accepted.

Author Response

Thank you for comments, we are glad to have addressed your comments.

Reviewer 3 Report

Line 6 should be “filter”?

Line 19 “recording” or “measuring” would be better than reading out

Line 54 “In addition to the focus on the ‘web’ “ rephrase maybe In addition to web inspired sensing structures

Line 64 “which combines”

Line 113. The variation in height is likely to be more significant than that in width.  I can’t find where you’ve addressed this? Also a 33% variation in width seems significant.

Line 101 Still unsure why the random number added to even spacing should not also be allowed to be negative 2pi()/r should give even spacing so if you only ever add a positive angle to this all the spacings should be larger than evenly spaced (with the exception of the last one which will have to be smaller). I suppose it works if you allow the addition of radii past 2pi() but then you’re skewing the randomness of the placement towards the 0 angle point. 

Line 104 Still no explanation of why these are considered dominant

Line 147

From the adxl335 datasheet: “Bandwidths can be selected to suit the application, with a range of 0.5 Hz to 1600 Hz for the X and Y axes, and a range of 0.5 Hz to 550 Hz for the Z axis.” 

Perhaps the paper title should be reconsidered if the objective has changed from computational design to design exploration

Author Response

Please see document attached.
